# Smoker, ex-smoker or non-smoker? The validity of routinely recorded smoking status in UK primary care: a cross-sectional study

Louise Marston,[1] James R Carpenter,[2,3] Kate R Walters,[1] Richard W Morris,[1] Irwin Nazareth,[1] Ian R White,[4] Irene Petersen[1]

▶ Prepublication history and additional material is available. To view please visit the journal (http://dx.doi.org/10.1136/bmjopen-2014-004958).

[1]Department of Primary Care and Population Health, University College London, London, UK
[2]Department of Medical Statistics, London School of Hygiene and Tropical Medicine, London, UK
[3]MRC Clinical Trials Unit, London, UK
[4]MRC Biostatistics Unit, Cambridge Institute of Public Health, University Forvie Site, Cambridge, UK

**Correspondence to**
Dr Louise Marston;
l.marston@ucl.ac.uk

## ABSTRACT

**Objective:** To investigate how smoking status is recorded in UK primary care; to evaluate whether appropriate multiple imputation (MI) of smoking status yields results consistent with health surveys.

**Setting:** UK primary care and a population survey conducted in the community.

**Participants:** We identified 354 204 patients aged 16 or over in The Health Improvement Network (THIN) primary care database registered with their general practice 2008–2009 and 15 102 individuals aged 16 or over in the Health Survey for England (HSE).

**Outcome measures:** Age-standardised and age-specific proportions of smokers, ex-smokers and non-smokers in THIN and the HSE before and after MI. Using information on time since quitting in the HSE, we estimated when ex-smokers are typically recorded as non-smokers in primary care records.

**Results:** In THIN, smoking status was recorded for 84% of patients within 1 year of registration. Of these, 28% were smokers (21% in the HSE). After MI of missing smoking data, the proportion of smokers was 25% (missing at random) and 20% (missing not at random). With increasing age, more were identified as ex-smokers in the HSE than THIN. It appears that those who quit before age 30 were less likely to be recorded as an ex-smoker in primary care than people who quit later.

**Conclusions:** Smoking status was relatively well recorded in primary care. Misclassification of ex-smokers as non-smokers is likely to occur in those quitting smoking at an early age and/or a long time ago. Those with no smoking status information are more likely to be ex-smokers or non-smokers than smokers.

## INTRODUCTION

A fifth of the British adult population is of smokers[1] and there is still a need for further research into smoking and smoking-related diseases including coronary heart disease (CHD) and stroke, respiratory diseases and cancers. Routinely collected smoking data

### Strengths and limitations of this study

- This study includes data from 'real'-life primary care electronic records.
- First study to compare the definition of smoking status in primary care with a population survey.
- Study focuses on data recorded in the first year after patient registration and may not be applicable to other times.

can be used in clinical practice to identify populations at risk of smoking-related diseases, such as identifying smokers to undergo spirometry testing for early diagnosis of chronic obstructive pulmonary disease (COPD), or to be invited for smoking cessation services. It is important to understand the accuracy of the data, and whether cases may be missed in those with no recorded smoking status. Electronic health records, including primary care databases, have proved to be very powerful resources for epidemiological and health research,[2–12] allowing research that would be difficult using primary research methods; for example, studying the elderly and people with severe mental illness.[4 7 9 11] Additionally, they include millions of patients giving power to study rare conditions. Nevertheless, as they are collected for clinical reasons, they raise a number of issues when used for research; not the least of these are missing data.

In order to conduct such research, it is important to understand how smoking status is recorded in primary care and how missing data may be addressed. There is evidence that the recording of smoking status has improved substantially in UK primary care[13 14] and that estimates of *current smoking* are similar to those in large population surveys.[15 16] Most general practices now routinely record smoking status at regular

intervals as a part of the Quality Outcome Framework.[17] However, we do not know how the different and non-standardised classifications of *ex-smokers and non-smokers* in primary care records compared with the standardised recording of smoking status in population surveys such as the Health Survey for England (HSE).

As noted already, a proportion of patients still lack a smoking status record in their primary care records. It is unclear how to deal with these patients when conducting research where smoking status is either the outcome of the research or an explanatory factor for patients' health.[3] [6] [18] [19] Methodological research has demonstrated that including only patients with complete records can substantially bias the results, especially when the reason for missing data is associated with patient outcomes.[20] [21] In recent years, efforts have been made to address missing data in primary care databases[3] [19] [22] using multiple imputation, though reporting on the comparability of the results of multiple imputation with population data has been sparse. Therefore, it is unclear whether multiple imputation accurately replicates data representing the population.[3] [6] [19] [23] Our previous work on missing data in The Health Improvement Network (THIN) primary care database showed that many health indicator measurements (eg, weight and blood pressure) recorded within the first year of patients' registration at a general practice were comparable with large external datasets before and after multiple imputation.[18] However, smoking status was not directly comparable with data from the HSE. Although the proportion of smokers was similar between THIN and the HSE *before* multiple imputation of data in THIN, the proportion of smokers was substantially higher *after* multiple imputation in THIN. On the other hand, the proportion of ex-smokers was substantially lower in THIN both *before* and *after* imputation compared to the HSE. This suggests that current smokers may be adequately identified using primary care data and that most people with missing data on smoking status are likely to be either ex-smokers or non-smokers. This has clinical importance as smoking status (including ex-smoking) may be used to identify those at risk of disease, for example, COPD or cardiovascular disease.

In this study, we further investigate recording of smoking status in primary care and explore potential reasons for the discrepancy in the proportion of ex-smokers between primary care records and the HSE. Specifically, we seek to deduce when ex-smokers may not be recorded as such in primary care records based on information about time since quitting in the HSE. Finally, we aim to provide a practical solution for imputation of missing smoking status records in routinely collected clinical data.

## METHODS
### Study populations
We used data from THIN primary care database, from practices in England that had passed data quality checks,

to ensure that they were using their computer system to record all patient consultations.[24–26] In the UK, 98% of the population are registered with a National Health Service (NHS) general practitioner (GP) to receive routine healthcare.[27] THIN is broadly representative of all general practices in the UK in terms of age and sex of patients, practice size and geographical distribution.[28] The database contains information on sociodemographics, symptoms, diagnoses, referrals to secondary care, prescribing, results of tests and health status indicators.

For this study, we selected patients aged 16 years or over who registered with a general practice between 1 January 2008 and 31 December 2009 (N=354 204) and were registered for at least a year. We examined records from the first year after the patient registered, hence using data up to the end of 2010. Many people have a 'new patient check' soon after registration, where information on demographics, health indicators and disease status is collected.

We compared the distribution of smoking status with that in the HSE from 2008 for those aged 16 years or over (N=15 102). The HSE is a national annual cross-sectional interview-based survey of approximately 22 000 people.[29] The survey includes questions on sociodemographics, general health and information on smoking status. The HSE has nearly complete records of smoking (99.3%), and we therefore used the data from patients with complete smoking information.

### Definition of smoking status
In THIN, smoking status was recorded by self-report. In many general practices, this would be on the basis of a questionnaire submitted at the time of registration, whereas in other general practices this would be recorded in conjunction with a clinical consultation with the GP or practice nurse. GPs and nurses may be more interested in the separation between current non-smokers and smokers; thus, the non-smoking categories may include some people who are never smokers as well as some who are ex-smokers in primary care records. In THIN, we extracted smoking status data either using Read codes,[30] which were classified into non-smoker, ex-smoker and smoker with clinical input, or we used the categorisation (non-smoker, ex-smoker or current smoker) provided in the Additional Health Data. In the HSE, smoking status was defined on the basis of a series of questions (see online supplementary appendix 1), and individuals who had ever smoked (but did not smoke at the time of the interview) would be defined as ex-smokers, regardless of their age at quitting and length of time since they quit. The HSE holds information on when ex-smokers quit so that age at the time they quit can be deduced, whereas this information was not consistently available in THIN.

### Statistical analyses
Initially, we examined smoking status (smoker, ex-smoker, non-smoker or missing) in THIN and the

HSE, overall, by age group, gender and Index of Multiple Deprivation 2004 (IMD) quintile.[31] Then we used multiple imputation to impute missing data in THIN. Multiple imputation via full conditional specification was performed using Stata's 'ice' command.[32 33] Multiple imputation is a statistical method which uses the data available to model the likely distribution of missing data.[20] A number of imputed datasets are produced in each of which plausible values are drawn from the imputation model. The method is designed to correctly reflect the uncertainty surrounding the missing values. With an appropriate imputation model, multiple imputation is an unbiased method of accounting for missing data. It is usually performed under the missing at random (MAR) assumption, but it may also be performed under specific missing not at random (MNAR) assumptions. These methods have been described in greater detail elsewhere.[20 34–36]

After preliminary analysis,[34] we included the following variables in the multiple imputation models: age in years, gender and IMD quintile,[31] health indicators: smoking status (three categories, non-smoker, ex-smoker and current smoker), height, weight, systolic and diastolic blood pressures and disease indicators: type II diabetes, CHD and cerebrovascular accident. There were missing values for smoking status, blood pressure, weight, height and IMD quintile. Within the full conditional specification imputation algorithm, continuous variables were imputed using multiple linear regression, smoking status using multinomial regression and IMD quintile using ordered logistic regression. Percentages in each smoking category were obtained using Rubin's rules.[37] In the first multiple imputation, we assumed that smoking data were MAR and hence allowed imputed smoking data of smokers, non-smokers or ex-smokers (using an MAR assumption; hereafter referred to as MAR MI). In the second multiple imputation, we assumed that all smokers had been recorded (so that smoking data were MNAR) and we imputed missing smoking data as either ex-smokers or non-smokers (hereafter referred to as MNAR MI).

Following multiple imputation, we carried out age-specific direct standardisation using the HSE as the standard population and the age-specific proportion in each smoking category from THIN. This was carried out to account for the fact that the mean age in the HSE was 49 years while the mean age in THIN was 38 years in the year after registration.

We deduced the average time after which an ex-smoker is no longer classified as an ex-smoker in primary care records by combining information from the HSE on when ex-smokers quit and the age-specific distribution of ex-smokers in THIN after imputation of non and ex-smokers. This was carried out by ranking the individuals in the HSE in accordance to the length of time since they quit by 10-year age groups and then 'reclassifying' individuals who had quit the longest time ago within each age group from ex-smoker to non-smoker until we reached the same proportion of ex-smokers in the HSE as in THIN. By doing this, we were able to estimate the average time that elapses from quitting smoking after which true ex-smokers are recorded as non-smokers in primary care records.

## RESULTS

In total, 354 204 individuals were included from 366 general practices in THIN and 15 102 individuals from the HSE. Individuals in THIN were, on average, 11 years younger than those in the HSE (38 vs 49 years, respectively; table 1). Smoking status was recorded for 84% in THIN within 1 year of initial registration. Before multiple imputation of missing data, a greater proportion of people were recorded as smokers in THIN than in the HSE (24% vs 21%, respectively), and the proportions of ex-smokers and non-smokers differed substantially between THIN and the HSE (table 1).

Our first analyses used missing as a separate category of smoking, so we refer to those with reported smoking status as 'known smokers' and 'known ex-smokers'. The proportion of known smokers by age group was similar in THIN and the HSE between 30 and 79 years, but this was not the case for the proportions of known ex-smokers and non-smokers (figure 1). In the HSE, the proportion of ex-smokers increased from 12% within the 20–29 age group to 46% in the 80–89 age group. In THIN, the proportion of known ex-smokers also increased with age, although the overall proportion of known ex-smokers was smaller than in the HSE for all age groups after 20–29 years. Conversely, in the HSE, the proportion of non-smokers decreased slightly from 56% in the 20–29 age

**Table 1** Summary statistics for THIN in the first year of registration and the HSE 2008

| Variable | THIN | | HSE | |
|---|---|---|---|---|
| | N | Per cent | N | Per cent |
| Male | 164 085 | 46 | 6760 | 45 |
| Female | 190 119 | 54 | 8342 | 55 |
| Missing sex | | 0 | | 0 |
| Non-smoker | 165 618 | 47 | 7874 | 52 |
| Ex-smoker | 49 874 | 14 | 3966 | 26 |
| Current smoker | 83 526 | 24 | 3158 | 21 |
| Missing smoking status | 55 186 | 16 | 104 | 1 |
| Age years mean (SD) | 38 | (17) | 49 | (19) |
| Missing age | | 0 | | 0 |
| Least deprived | 69 104 | 20 | 3321 | 22 |
| Quintile 2 | 71 771 | 20 | 3039 | 20 |
| Quintile 3 | 66 422 | 19 | 3010 | 20 |
| Quintile 4 | 71 789 | 20 | 2928 | 19 |
| Most deprived | 52 120 | 15 | 2804 | 19 |
| Missing IMD | 22 998 | 6 | 0 | 0 |

HSE, Health Survey for England 2008; IMD, Index of Multiple Deprivation; THIN, The Health Improvement Network.

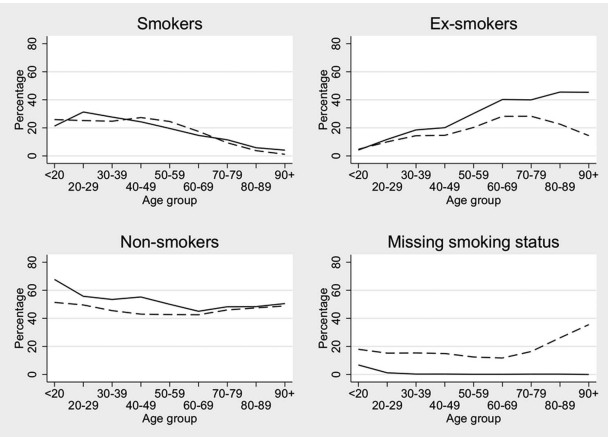

**Figure 1** Smoking status percentages in The Health Improvement Network (THIN) and the Health Survey for England (HSE) 2008 by age group. Solid line is the HSE 2008; dashed line is THIN.

group to 48% in the 80–89 age group. Within THIN, the proportion of known non-smokers remained constant with increasing age at around 43%. The proportion of missing smoking data in THIN was relatively constant at less than 20% until the 70–79 years age group, but increased substantially thereafter (figure 1).

In THIN, the percentage of non-smokers was greater for women (52%) than men (40%) while the percentage of known smokers was smaller for women (21%) than men (27%). There were similar trends in the HSE, although the percentage differences between sexes were smaller (smokers: 22% of men vs 20% of women).

The proportions in each smoking status category varied substantially by social deprivation in both THIN and the HSE (figure 2). In THIN, the percentage of non-smokers decreased from 52% in the least deprived

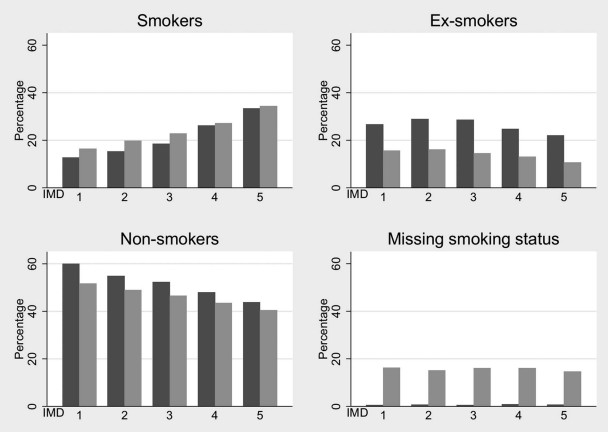

**Figure 2** Smoking status percentages in The Health Improvement Network (THIN) and the Health Survey for England (HSE) 2008 by deprivation quintile. Index of Multiple Deprivation (IMD) 1 is the least deprived and IMD 5 is the most deprived. Darker bars represent the HSE 2008 and lighter bars represent THIN.

quintile to 40% in the most deprived quintile. The percentage of known ex-smokers decreased slightly with increasing deprivation. In contrast, the percentage of known smokers increased with increasing deprivation from 16% in the least deprived quintile to 34% in the most deprived quintile (figure 2). The patterns were similar in the HSE, although the proportion of ex-smokers was substantially larger across all levels of deprivation in the HSE compared to THIN.

### Analyses imputing missing smoking status

After MAR MI of THIN, age-standardised smoking prevalences still differed somewhat between THIN and the HSE. For example, 22% were ex-smokers in THIN compared with 26% in the HSE; 25% were smokers in THIN, compared with 21% in the HSE (table 2).

After MNAR MI of THIN (ie, specifying that missing values are either ex-smokers or non-smokers), the age-standardised prevalence of smoking in THIN was similar to that in the HSE (table 2). However, the age-specific prevalence of ex-smokers was still greater in the HSE than in THIN. Age-specific analysis showed that this difference was greatest at older ages, and indeed reversed at younger ages. This suggested that individuals who had quit in the less recent past might be classified as non-smokers in THIN but as ex-smokers in the HSE (figure 3).

The median time since ex-smokers quit in the HSE varied greatly by age group (table 3), from 2 years (IQR: 0, 3) in the under 20–40 (IQR: 25, 51) years in those aged 90 or over (table 3). Equating proportions of ex-smokers in THIN to that in the HSE data suggested that the typical time-window after which patients are no longer regarded as ex-smokers in primary care, but instead regarded as non-smokers, varied with age. Thus, typically, individuals who registered with a general practice when they were in their 40s would no longer be recorded as an ex-smoker if they quit more than 22 years earlier (when they were between 18 and 27 years of age; table 3). Individuals registering in their seventies would typically no longer be recorded as ex-smokers if they quit 42 years earlier (when they were between the ages of 28 and 37 years; (table 3). Yet, most individuals who quit after the age of 30 would still be captured as ex-smokers when they later registered with a new general practice. Using these age-specific extrapolations to reclassify ex-smokers as non-smokers in the HSE according to when they quit, we can see that the age-specific distributions of ex-smokers in THIN and the reclassified HSE are similar (figure 3).

### DISCUSSION

The proportion of newly registered patients in THIN between 2008 and 2009 with a record of being a smoker was slightly higher than that in the HSE in 2008. However, the proportion of individuals recorded as ex-smokers and non-smokers differed substantially

**Table 2** Percentages within each smoking status for THIN and the HSE 2008 after various adjustments

| Category | THIN | | | HSE | |
| --- | --- | --- | --- | --- | --- |
| | Complete records (%) | After MAR MI (%)*† | After MNAR MI (%)*‡ | Observed (%) | Reclassifying ex-smokers (%)§ |
| Non-smoker | 55 | 53 | 57 | 53 | 57 |
| Ex-smoker | 17 | 22 | 23 | 26 | 22 |
| Smoker | 28 | 25 | 20 | 21 | 21 |

*Directly standardised using the HSE age distribution as standard.
†Imputed assuming that missing values are smokers, non-smokers or ex-smokers.
‡Imputed assuming that missing values are non-smokers or ex-smokers.
§Within each age group, reclassifying the optimum number of ex-smokers as non-smokers based on the distributions shown after MNAR MI.
HSE, Health Survey for England 2008; MAR, missing at random; MI, multiple imputation; MNAR, missing not at random; THIN, The Health Improvement Network.

between THIN and the HSE. Overall, a larger proportion of individuals were recorded as ex-smokers in the HSE than in THIN and this increased with age. Likewise, the proportion of ex-smokers was substantially larger across all levels of deprivation in the HSE compared to THIN.

Under MAR MI, there was a greater percentage of smokers (25%) and a smaller percentage of ex-smokers (22%) in THIN compared with the HSE (smokers 21%, ex-smokers 26%). However, MNAR MI (assuming all missing data were either ex-smokers or non-smokers) slightly increased the proportion of non-smokers (57%) in THIN compared to the HSE (53%), whereas the proportion of ex-smokers (23%) was slightly lower in THIN. Moreover, the latter imputation resulted in a relatively larger percentage of ex-smokers in THIN in those aged under 30 years compared with the HSE. This may be because the imputation model was unable to distinguish between ex-smokers and non-smokers in those age groups as both are unlikely to have developed typical later onset diseases which are key predictors of smoking status in the imputation model.

There may be several reasons for the discrepancy in the distribution of the smoking categories between THIN and the HSE. In the HSE, the definition of an ex-smoker was highly sensitive and clearly defined.[29] Thus, respondents were categorised as ex-smokers even if they were a trivial smoker, smoked for a short period of time and/or quit many decades ago. Also, the HSE used computer-aided personal interviewing, where questions were read to the respondent in a standardised way from the screen and a detailed sequence of questions were asked to ascertain current smoking status. In primary care, while smoking status is systematically recorded in medical records, there is no detailed protocol for recording smoking status and the ascertainment is thus likely to vary by how the information was obtained. Many practices use self-report questionnaires at registration including smoking status. Smoking status is then updated by health professionals (GPs and/or practice nurses) during consultations where smoking status is often recorded as part of an assessment of current or future disease risk.

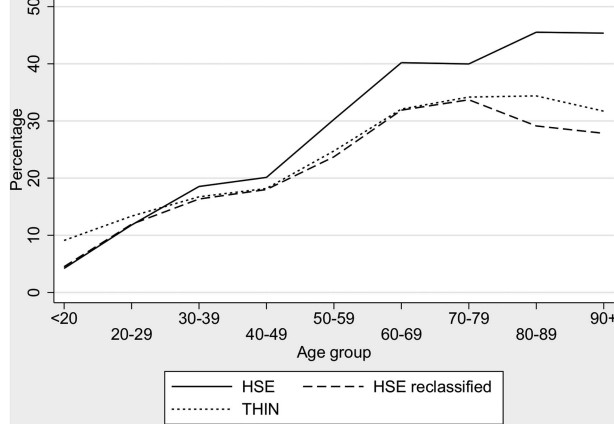

**Figure 3** Age group-specific percentages of ex-smokers in The Health Improvement Network(THIN; after MNAR (missing not at random) imputation) and the Health Survey for England (HSE) 2008 (before and after reclassifying ex-smokers in the HSE who quit before the age specified in table 3 column 3 to be non-smokers).

**Table 3** Age-specific percentiles of time since quitting smoking in the Health Survey for England 2008

| Age group | Median time since quitting (years) | Extrapolated number of years since quitting | Extrapolated age when they quit |
| --- | --- | --- | --- |
| <20 | 2 | * | * |
| 20–29 | 3 | * | * |
| 30–39 | 5 | 14 | 16—25 |
| 40–49 | 10 | 22 | 18—27 |
| 50–59 | 20 | 30 | 20—29 |
| 60–69 | 24 | 35 | 25—34 |
| 70–79 | 30 | 42 | 28—37 |
| 80–89 | 32 | 40 | 40—49 |
| 90+ | 40 | 46 | 44+ |

*Not possible to assign an optimal value for reclassification to these age groups.

Our examination of the age-standardised data suggests that typically an ex-smoker in primary care settings is recorded as a non-smoker when they quit at a young age or had not smoked for a substantial time period. This could be because the patient may not volunteer previous smoking in either in the initial self-report questionnaire or on questioning by clinicians when it was minor, long ago or they considered it not relevant to their current or future health. It is possible that patients are more reluctant to volunteer ex-smoking habits when data are being held on their medical record and are not anonymous. However, comparing the proportion of individuals with a smoking record in THIN with that of the HSE, we found a similar distribution suggesting that most smokers were identified in the first year of their registration in primary care. Similar findings have been observed in the literature by calendar year.[18] With the introduction of the Quality and Outcomes Framework in 2004, there has also been increased incentive to identify smokers in relation to specific disease outcomes.[38 39] Indeed, we found in our previous study that those with respiratory and cardiac conditions were more likely to have any smoking status recorded within the first year of registration.[13] Smoking status was validated in the HSE in 2007 by the use of saliva cotinine samples and was found to be accurate.[40]

The method of age standardisation then deducing the average time since quitting and reclassifying them to non-smokers in the HSE is relatively crude and assumes that everyone who becomes an ex-smoker does so at the same time in their lives as others in their age group. However, it is likely to be indicative of reporting of smoking status at the GP practice, given the results shown in this study.

An alternative method of dealing with unobserved smoking data is to dichotomise smoking status into current smokers and non-current smokers with missing data assumed to pertain to non-current smokers. However, it should be noted that this solution may be to the detriment of some epidemiological studies where ex-smokers who quit recently are at greater risk of disease than non-smokers. For example, the 50-year follow-up of male British doctors shows that ex-smokers had elevated age standardised mortality rates for many diseases.[41 42]

Our findings suggest that *in contrast* to health surveys, patients who quit smoking at a young age (before 25–30) are likely to be recorded by their general practice as a non-smoker instead of an ex-smoker. This has implications for researchers using these data sources. To the best of our knowledge, this is the first study which seeks to deduce and quantify typical time between when a smoker quit and when they are no longer perceived as an ex-smoker in primary care. Clinicians, policy-makers and researchers who wish to use smoking status in primary care records to identify populations at risk of smoking-related diseases can be reassured by our findings that by using data from new registrations, most current smokers will be identified and misclassification of ex-smokers is more likely to have occurred in those who have quit smoking at an early age and/or a long time ago.

**Contributors** LM extracted and analysed the data and wrote the first draft of the paper with help from IP and JRC. KRW and IN provided clinical input and IRW and RWM provided additional statistical input. All authors commented on the paper and helped write subsequent drafts.

**Funding** This study is funded by a UK Medical Research Council grant [G0900701].

**Competing interests** JRC was funded by a UK Economic and Social Research Council research fellowship grant [RES-063-27-0257]. IRW was funded by a UK Medical Research Council grant [U105260558].

**Ethics approval** The data provider (CSD-MR) obtained overall ethical approval from the South East MREC (MREC/03/01/073) and this study was further approved by The Health Improvement Network (THIN) scientific review committee.

**Provenance and peer review** Not commissioned; externally peer reviewed.

**Data sharing statement** No additional data are available.

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
