## [Reviewer comments · BMJ Open]

Some articles will have been accepted based in part or entirely on reviews undertaken for other BMJ Group journals. These will be reproduced where possible.

ARTICLE DETAILS

TITLE (PROVISIONAL)	Smoker, ex-smoker or non-smoker? The validity of routinely recorded smoking status in UK primary care: a cross-sectional study
AUTHORS	Marston, Louise; Carpenter, James; Walters, Kate; Morris, Richard; Nazareth, Irwin; White, Ian; Petersen, Irene

VERSION 1 - REVIEW

REVIEWER	David Tappin University of Glasgow UK
REVIEW RETURNED	20-Feb-2014

GENERAL COMMENTS	I found the main issue with this paper was how to gauge the importance of using this routinely collected data. I think if the authors provided specific instances when the data was used in research and how the research findings had been improved by imputation. I am sure the authors have thought of this, but is there a way to randomly take items of smoking data for individuals away from a dataset and then just replace those items of data for those individuals using multiple imputation. This would allow a reader to judge if multiple imputation gave an accurate overall prevalence of smoking and which type of imputation should be used.
--

REVIEWER	Dr Lisa Szatkowski University of Nottingham, UK
	I have published work in the same area: Szatkowski et al. J Epidemiol Community Health 2012;66:791-5. Langley et al. BMC Public Health 2011;11:773.
REVIEW RETURNED	24-Feb-2014

GENERAL COMMENTS	The manuscript is well written and addresses an important question. I do, however, have a number of questions and suggestions for improvements: - The introduction makes a good case for the current study. However, this work builds on work published by myself and colleagues which in my opinion should be cited in the introduction: Szatkowski et al, JECH, 2012, 66:791-795. Langley et al, BMC Public Health, 2011, 11:773.- You state that "we do not know how the different and non-standardised classifications of ex, non and current smokers in primary care records compare to the standardised recording of smoking status in population surveys such as the Health Survey for
---

	England” (p5 line 40). This statement is not entirely true – the studies by Szatkowski and Langley mentioned above compare recording in primary care to data from the General Lifestyle Survey.  - I would suggest the use of the term ‘never smoker’ throughout rather than ‘non smoker’ if this is indeed what you mean. Non smoker can easily be interpreted as someone who doesn’t smoke at the point they are asked, but who may have smoked at some point in the past. Indeed, the HSE uses the terminology never and ex smokers, and my experience with the coding of smoking status in THIN using Read codes suggests that the specific codes for non smokers are used very infrequently – the explicit code for never smoker is used far more frequently. - These results are now a few years’ old – are you able to update using more recent THIN data? - Please state how many practices your 354,204 patients came from. - Did you require patients to remain registered for the whole year of follow-up, without death or transfer out of the practice? Please clarify. - Did you use AMR date as a restriction to ensure you only included data from practices once they had reached this data quality date? Please add details to the manuscript as appropriate. - How did you account for the complex survey design of the HSE? - Did you just use THIN data from English practices or those elsewhere in the UK too? For comparison with the HSE, a restriction to THIN data from English practices only would be most appropriate, given the known variations in smoking prevalence between different parts of the UK. If you’ve used THIN data from the whole of the UK then comparison with the GLF may be more appropriate, though that would of course increase the overlap with the Szatkowski/Langley work. - Please explain how you used the Read codes recorded in THIN to derive the patients’ smoking status. - P8 last paragraph – “Patients would be classed as current non-smoker, or current smokers. In some instance the non-smokers would be classified as ex-smokers but this was variably defined from one practice to another.” Is this statement based on assessment of the THIN data, anecdote, conversation with GPs, or some other evidence? Please justify your reasoning. - On p9 line 30 you state that you used MI to impute missing smoking status, and then on p10 line 18 mention how continuous outcomes, smoking status and IMD were imputed. Please clarify exactly which variables you imputed. - In Table 1 please show the breakdown by IMD including the amount of missing data - How complete were the recordings of height, weight, blood pressure, diabetes, CHD and CVA? - P18 line 32 – you allude to having analysed underreporting by pregnancy status, but there is no mention of this earlier in the manuscript. If you wish to retain this sentence please provide details of your analysis in the methods. However, I don’t think it is surprising that you found no excess underreporting in pregnancy – your study population is newly registered patients for whom practices are expected to record smoking status within 3 months under the QOF. The fact that women may be pregnant is almost incidental.
--	---

REVIEWER	Louisa Jorm University of Western Sydney Australia
REVIEW RETURNED	26-Feb-2014

GENERAL COMMENTS	I have a few suggestions to improve the paper: (i) This detailed analysis extends the author's earlier work using similar datasets (Marston L, Carpenter JR, Walters KR et al. Issues in multiple imputation of missing data for large general practice clinical databases. Pharmacoepidemiology and Drug Safety 2010; 19: 618–626, reference 16) which demonstrated that missing smoking status in THINS was like to be missing not at random (MNAR). However, there seem to be some differences between the current findings regarding smoking status in THINS and the earlier ones - for example fewer missing values, and a smaller impact of missing at random (MAR) multiple imputation (MI) on the prevalence of current smoking. It would be useful to include some commentary on the differences between the two analyses in the Discussion. Do they reflect temporal trends in the completeness and quality of THINS data? (ii) The methods used to "reclassify" ex-smokers in the HSE from "ex-" to "non-" based on the age-specific proportion of ex-smokers in THIN after MI was a little bit hard to follow and I am not convinced of the value of this exercise. Did this use the results of MAR or MNAR MI? From Figure 3, it looks like the MNAR results were used. Given the method used to "reclassify", is it not to be expected that the resulting distribution of ex- and non-smokers would be virtually identical to that in THIN after the relevant MI? This approach seems a bit circular. I would suggest that some additional justification of this method, and discussion of the practical implications of its findings, be added. Also I think the wording of footnote (d) in Table 2 could be improved - by specifying which MI method was used to reclassify and perhaps by making reference to Table 3 as is done in Figure 3. (iii) The wording of the following sentence (page 10, paragraph 1) could be improved: "Multiple imputation was performed using Chained Equations using the ice command using Stata"
---

REVIEWER	Robert Stewart King's College London, UK
REVIEW RETURNED	04-Mar-2014

GENERAL COMMENTS	The study's findings rest on a comparison between THIN-derived data and findings from the HSE. There seems to be an implicit assumption that the HSE data are therefore 'gold standard' and it might therefore be helpful to have a couple of sentences in the Methods or Discussion considering this - in particular, the likely community representativeness of the HSE samples. I think it would be helpful to have a review by a specialist in multiple imputation prior to publication as I didn't feel I had sufficient expertise to judge this aspect.
---

VERSION 1 – AUTHOR RESPONSE

Reviewer Name David Tappin

I found the main issue with this paper was how to gauge the importance of using this routinely collected data. I think if the authors provided specific instances when the data was used in research and how the research findings had been improved by imputation.

We have added to the introduction, paragraph 1 “Electronic health records, including primary care databases, have proved to be very powerful resources for epidemiological and health research.[2-12], allowing research that would be difficult using primary research methods; for example, studying the elderly and people with severe mental illness.[4, 7, 9, 11] Additionally they include millions of patients giving power to study rare conditions. Nevertheless, as they are collected for clinical reasons, they raise a number of issues when used for research; not least of these is missing data.”.

However, if studies are done in only those with complete records the study sample may be substantially reduced and the results may be biased (Sterne et al, 2009).

Sterne JAC, White IR, Carlin JB et al. Multiple imputation for missing data in epidemiological and clinical research: potential and pitfalls. Br Med J 2009;b2393

I am sure the authors have thought of this, but is there a way to randomly take items of smoking data for individuals away from a dataset and then just replace those items of data for those individuals using multiple imputation. This would allow a reader to judge if multiple imputation gave an accurate overall prevalence of smoking and which type of imputation should be used.

Yes, if we remove data under a specific known mechanism, we can indeed verify the statistical properties of the multiply imputed data are as they should be. The algorithm we used in this work, and the specific programme (multiple imputation), has been verified in this way. Hence, when we apply this algorithm to THIN, we can reasonably deduce that the missing data mechanism that gives best match to the HSE data is likely to represent what is actually happening.

Reviewer Name Dr Lisa Szatkowski

The introduction makes a good case for the current study. However, this work builds on work published by myself and colleagues which in my opinion should be cited in the introduction: Szatkowski et al, JECH, 2012, 66:791-795. Langley et al, BMC Public Health, 2011, 11:773.

Thank you for highlighting these papers, we have now cited them in the introduction.

You state that “we do not know how the different and non-standardised classifications of ex, non and current smokers in primary care records compare to the standardised recording of smoking status in population surveys such as the Health Survey for England” (p5 line 40). This statement is not entirely true – the studies by Szatkowski and Langley mentioned above compare recording in primary care to data from the General Lifestyle Survey.

We have changed the sentence in question as Szatkowski and colleagues have addressed the issue of percentage current smokers: “However, we do not know how the different and non-standardised classifications of ex and, non-smokers in primary care records compared to the standardised recording of smoking status in population surveys such as the Health Survey for England (HSE).”

I would suggest the use of the term ‘never smoker’ throughout rather than ‘non smoker’ if this is indeed what you mean. Non smoker can easily be interpreted as someone who doesn’t smoke at the

point they are asked, but who may have smoked at some point in the past. Indeed, the HSE uses the terminology never and ex smokers, and my experience with the coding of smoking status in THIN using Read codes suggests that the specific codes for non smokers are used very infrequently – the explicit code for never smoker is used far more frequently.

We feel that it is appropriate to maintain the term non-smoker, as we believe that some people who are recorded as not smoking in THIN are actually ex-smokers. That is, the GP is correct in coding them as non-smokers, but some GPs may also use this term for never smokers. However, we can be more certain that those who are non-smokers in the Health Survey for England are actually never smokers as they use a strict algorithm to derive smoking status. Therefore we feel to use non-smoker for those in THIN and never smoker for those in the HSE may be more confusing to the reader. Indeed, one of the aims of this study was to disentangle the proportion of those with non-smoking record in primary care are true never smokers.

We have added further clarification to the Definition of smoking status section “GPs and nurses may be more interested in the separation between current non-smokers and smokers, thus the non-smoking categories may include some people who are never smokers as well as some who are ex-smokers in primary care records.”

These results are now a few years' old – are you able to update using more recent THIN data?

We are not able to update these to a more recent version as the data are contemporaneous with the HSE data, which is one which focuses on cardiovascular risk factors. The focus of this study is primarily on the recording of smoking status, not time trends. Additionally, when doing a longitudinal research study, many of the baseline health indicators would come from this time period.

Please state how many practices your 354,204 patients came from.

Data come from 366 general practices. This has been included in the first line of the results on page 11. “In total, 354,204 individuals were included from 366 general practices in THIN and 15,102 individuals from the HSE”

Did you require patients to remain registered for the whole year of follow-up, without death or transfer out of the practice? Please clarify.

Patients had to be registered at the practice for a whole year of follow up. This has been included on page 8. “For this study we selected patients aged 16 years or over who registered with a general practice between 1st January 2008 and 31st December 2009 (N=354,204) and were registered for at least a year.”

Did you use AMR date as a restriction to ensure you only included data from practices once they had reached this data quality date? Please add details to the manuscript as appropriate.

Yes we did, as all practices included met AMR criteria by 01/01/2008. We have included some more explanation in the Study Population section: “We used data from THIN primary care database, from practices in England that had passed data quality checks, to ensure they were using their computer system to record all patient consultations.”

How did you account for the complex survey design of the HSE?

Our comparison between the HSE and THIN was age and sex standardised as the population in the HSE was much older than those registered with a THIN practice included between 01/01/2008 and

31/12/2009. We have looked at the 2008 Health Survey for England report and found that women were over represented and men under 35 underrepresented. This pattern is also seen in THIN.

<http://www.hscic.gov.uk/catalogue/PUB00430/heal-surv-phys-acti-fitn-eng-2008-rep-v3.pdf>
Craig R, Mindell J, Hirani V (eds) (2009) Health Survey for England 2008, London: The Information Centre.

Did you just use THIN data from English practices or those elsewhere in the UK too? For comparison with the HSE, a restriction to THIN data from English practices only would be most appropriate, given the known variations in smoking prevalence between different parts of the UK. If you've used THIN data from the whole of the UK then comparison with the GLF may be more appropriate, though that would of course increase the overlap with the Szatkowski/Langley work.

The use of the deprivation variable was decided at the outset of the study. As the only deprivation variable available in the HSE was Index of Multiple Deprivation, we requested the IMD quintile from CSD-MR (the data provider). As IMD is only available in England, we only used English practices. We have included this in the Study Population section on page 7 "We used data from THIN primary care database, from practices in England that had passed data quality checks, to ensure they were using their computer system to record all patient consultations."

Please explain how you used the Read codes recorded in THIN to derive the patients' smoking status.

Read codes relating to any smoking status or treatment for smoking cessation were extracted from the Read code dictionary. These were then examined by one of the clinicians on the study (KW) to check whether any of the codes extracted did not relate to smoking status. With the remaining codes, we classified the codes into three categories. For example, Read codes starting 745H – smoking cessation therapies were coded as smokers as people would not be using smoking cessation therapy if they were not smokers and 137K000, Recently stopped smoking was classified as ex-smoker. We have added "In THIN we extracted smoking status data either using Read codes[29] which were classified into non-smoker, ex-smoker and smoker with clinical input, or we used the categorisation (non-smoker, ex-smoker or current smoker) provided in the Additional Health Data" to page 9.

P8 last paragraph – "Patients would be classed as current non-smoker, or current smokers. In some instance the non-smokers would be classified as ex-smokers but this was variably defined from one practice to another." Is this statement based on assessment of the THIN data, anecdote, conversation with GPs, or some other evidence? Please justify your reasoning.

We have added "GPs and nurses may be more interested in the separation between current non-smokers and smokers, thus some ex-smokers may be classified as non-smokers rather than ex-smokers." to page 9.

On p9 line 30 you state that you used MI to impute missing smoking status, and then on p10 line 18 mention how continuous outcomes, smoking status and IMD were imputed. Please clarify exactly which variables you imputed.

All variables with missing data were imputed, however, the focus of this paper is the imputation of smoking status, and other variables included in the imputation models were to aid the imputation of smoking status. We have made this more explicit on pages 9 and 10 by changing the sentence on page 9 to: "Then we used multiple imputation to impute missing data THIN" and adding a sentence on page 10: "There were missing values for smoking status, blood pressure, weight, height and IMD quintile."

In Table 1 please show the breakdown by IMD including the amount of missing data

IMD for THIN and the HSE has been added to Table 1.

How complete were the recordings of height, weight, blood pressure, diabetes, CHD and CVA?

In THIN recordings of diabetes, CHD and CVA were complete as we assume if there is not a Read code for these conditions in a patient's medical records, we assumed that they did not have the condition in question. Blood pressure data were present for 64% of patients, height for 65% and weight for 69%. We have not included this information in the current paper as we did a paper specifically on levels of missingness in health indicators (Marston et al, 2010), this information is not relevant in the current study.

P18 line 32 – you allude to having analysed underreporting by pregnancy status, but there is no mention of this earlier in the manuscript. If you wish to retain this sentence please provide details of your analysis in the methods. However, I don't think it is surprising that you found no excess underreporting in pregnancy – your study population is newly registered patients for whom practices are expected to record smoking status within 3 months under the QOF. The fact that women may be pregnant is almost incidental.

We have removed this sentence and reference from the reference list.

Reviewer Name Louisa Jorm

(i) This detailed analysis extends the author's earlier work using similar datasets (Marston L, Carpenter JR, Walters KR et al. Issues in multiple imputation of missing data for large general practice clinical databases. *Pharmacoepidemiology and Drug Safety* 2010; 19: 618–626, reference 16) which demonstrated that missing smoking status in THIN was like to be missing not at random (MNAR). However, there seem to be some differences between the current findings regarding smoking status in THIN and the earlier ones - for example fewer missing values, and a smaller impact of missing at random (MAR) multiple imputation (MI) on the prevalence of current smoking. It would be useful to include some commentary on the differences between the two analyses in the Discussion. Do they reflect temporal trends in the completeness and quality of THIN data?

Thank you for highlighting the differences between the data in the two papers. When using multiple imputation under the MAR assumption in our earlier paper (Marston et al, 2010) we found 55% were non-smokers, 16% were ex-smokers and 29% were smokers. In the present study, we found a slightly different distribution. These differences may be due to:

1 Random variation

2 Smoking prevalences decreasing over time. Our 2010 paper used data from 2006-2007 and the current paper used data from 2008-2009.

3 Our first paper using the whole of the United Kingdom. Smoking prevalences are different in the other constituent countries of the United Kingdom to England.

If the editor wishes, we would be happy to highlight the differences in the discussion of our paper.

(ii) The methods used to "reclassify" ex-smokers in the HSE from "ex-" to "non-" based on the age-specific proportion of ex-smokers in THIN after MI was a little bit hard to follow and I am not convinced of the value of this exercise. Did this use the results of MAR or MNAR MI? From Figure 3, it looks like the MNAR results were used. Given the method used to "reclassify", is it not to be expected that the resulting distribution of ex- and non-smokers would be virtually identical to that in THIN after

the relevant MI? This approach seems a bit circular. I would suggest that some additional justification of this method, and discussion of the practical implications of its findings, be added. Also I think the wording of footnote (d) in Table 2 could be improved - by specifying which MI method was used to reclassify and perhaps by making reference to Table 3 as is done in Figure 3.

We did the reclassification exercise with the HSE data because we wanted to understand why there were fewer ex-smokers in THIN than the HSE. We also wanted to identify the age that people may have quit before they are considered as ex-smokers in general practice records. By age specifically reclassifying those who quit the longest time ago, we were able to estimate the percentage of people in THIN who may be recorded as being a non-smoker, when in fact they should have been classified as an ex-smoker. We have added "By doing this, we were able to estimate the average time that elapses from quitting smoking after which true ex-smokers are recorded as non-smokers in primary care records." to page 11.

We have added to footnote d, Table 2: "Within each age group, reclassifying the optimum number of ex-smokers as non-smokers based on the distributions shown after MNAR MI".

(iii) The wording of the following sentence (page 10, paragraph 1) could be improved: "Multiple imputation was performed using Chained Equations using the ice command using Stata"

We have reworded this sentence to "Multiple imputation via full conditional specification was performed using Stata's "ice" command. [29, 30]" and moved it to the previous paragraph.

Reviewer Name Robert Stewart

The study's findings rest on a comparison between THIN-derived data and findings from the HSE. There seems to be an implicit assumption that the HSE data are therefore 'gold standard' and it might therefore be helpful to have a couple of sentences in the Methods or Discussion considering this - in particular, the likely community representativeness of the HSE samples.

The HSE was used because it is a large population based dataset, comparable in location and one that collected comparable data to the THIN dataset we used. It is also one of the datasets the UK Government cites when it publicises the percentage of people who are smokers. We therefore felt it would be helpful to make our comparison to the HSE. However, we discovered a large discrepancy in the way ex-smokers were identified in the two data sources. Therefore, we would be reluctant to say that the HSE is a "gold standard".

VERSION 2 – REVIEW

REVIEWER	Dr Lisa Szatkowski University of Nottingham, Division of Epidemiology and Public Health, UK
REVIEW RETURNED	04-Apr-2014

GENERAL COMMENTS	The authors have addressed my previous questions and suggestions and I have no further comments to make.
--

REVIEWER	Louisa Jorm University of Western Sydney, Australia
REVIEW RETURNED	02-Apr-2014

GENERAL COMMENTS

I would like to see some additional discussion of the practical implications of the findings for use of general practice smoking status data by other researchers - for example do the authors recommend that other researchers adopt MI methods, or conduct sensitivity analyses by reclassifying a proportion of non-smokers as ex-smokers?